# Lignin-Containing Coatings for Packaging Materials—Pilot Trials

**DOI:** 10.3390/polym13101595

**Published:** 2021-05-15

**Authors:** Asif Javed, Peter Rättö, Lars Järnström, Henrik Ullsten

**Affiliations:** 1Department of Engineering and Chemical Sciences, Karlstad University, SE-651 88 Karlstad, Sweden; peter.ratto@ri.se (P.R.); lars.jarnstrom@kau.se (L.J.); 2RISE Research Institutes of Sweden, Bioeconomy and Health, Biorefinery and Energy, Box 5604, SE-114 86 Stockholm, Sweden; 3Calamo AB, Box 6, SE-655 02 Molkom, Sweden; henrik.ullsten@calamo.se

**Keywords:** barrier coatings, glycerol, lignin, starch

## Abstract

One severe weakness of most biopolymers, in terms of their use as packaging materials, is their relatively high solubility in water. The addition of kraft lignin to starch coating formulations has been shown to reduce the water solubility of starch in dry coatings. However, lignin may also migrate into aqueous solutions. For this paper, kraft lignin isolated using the LignoBoost process was used in order to examine the effect of pH level on the solubility of lignin with and without ammonium zirconium carbonate (AZC). Machine-glazed (MG) paper was coated in a pilot coating machine, with the moving substrate at high speed, and laboratory-coated samples were used as a reference when measuring defects (number of pinholes). Kraft lignin became soluble in water at lower pH levels when starch was added to the solution, due to the interactions between starch and lignin. This made it possible to lower the pH of the coating solutions, resulting in increased water stability of the dry samples; that is, the migration of lignin to the model liquids decreased when the pH of the coating solutions was reduced. No significant difference was observed in the water vapor transmission rate (WVTR) between high and low pH for the pilot-coated samples. The addition of AZC to the formulation reduced the migration of lignin from the coatings to the model liquids and led to an increase in the water contact angle, but also increased the number of pinholes in the pilot-coated samples.

## 1. Introduction

Lignin, a heterogeneous and thermoplastic biopolymer, is one of the major constituents of wood and is the most hydrophobic component of the plant cell [1]. Lignin is extracted from wood in the pulping process, which separates cellulose fibers. The glass transition temperature ($T_{g}$) of lignin is dependent on the plant species, the extraction process, and the moisture content, and it ranges from 90 to 170 °C [2]. The $T_{g}$ of lignin can be reduced by adding a plasticizer or polymer with a low $T_{g}$ value, which can enhance the thermoplastic behaviour of lignin or of a lignin-based polymeric material [2]. Generally, lignin is extracted from the wood by mechanical, chemical, or enzymatic methods. Kraft, soda, sulfite, organosolv, hydrolysis, enzymatic, and LignoBoost processes are examples of lignin-isolation processes typically used to extract lignin from wood [2,3,4,5]. The LignoBoost process has many advantages, compared to other processes; including, e.g., lower investment and operational costs, as well as high lignin yield [5].

Lignin has shown great potential for use as an antioxidant and antimicrobial agent in food packaging [6,7,8]. Polymer blends based on lignin, such as starch–lignin, alginate–lignin, poly(vinyl alcohol)–lignin, and polyethylene–lignin blends, have recently been studied for their possible use in food packaging and in biomedical industries [9,10,11,12]. These studies have considered non-homogeneous blends of lignin with other polymers, where two-phase systems or composites were formed. For instance, Teramoto et al. [13] studied the formation of homogeneous blends of synthetic polymers, such as poly(vinyl acetate) and organosolv lignin, in organic solvents. Johansson et al. [14] and Bhat et al. [15] have studied waterborne starch–lignin blends for use in food packaging applications. Zadeh et al. [16] studied the properties of biopolymeric films based on lignin and soy protein isolate. The surface properties of spin-coated films of technical lignin have been investigated by Borrega et al. [17].

Spiridon et al. [18] and Kaewtatip and Thongmee [10] have investigated the interactions between starch and lignin in starch-based composite materials by Fourier-transform infrared spectroscopy and showed obvious interactions among the hydroxyl, carbonyl, and ether groups of the starch and lignin components. It was also indicated that the water resistance and tensile strength of starch-based composites films could be increased with the addition of lignin [10,18]. The effects of interactions between lignin derivatives and starch have been identified by Richardson et al. [19]. Gruber and Konish [20] discovered the ability of starch to form inclusion complexes with hydrophobic moieties.

In a previous study [21], the stability of starch–lignin films (prepared from solutions of pH 10) in contact with three model liquids—deionized water with approximately neutral pH, alkaline buffer solution with a pH of 10, and food simulant B (3%, *w/v*, acetic acid solution) with a pH of 2.4—was evaluated. The stability in water of starch–lignin films was enhanced when ammonium zirconium carbonate (AZC) was added to the formulations. It was shown that the migration of lignin from the films in contact with food simulant B at low pH was much lower than that from the films in contact with deionized water or alkaline buffer solution at high pH levels. Increased migration from the films to the model liquids was also observed if the pH adjustment was done with sodium hydroxide, compared to ammonia. As ammonia evaporates during drying, this indicates that residual bases in the films increase migration. McAlpine [22], Yoon and Deng [23], and Song et al. [24] have suggested that pH levels between 7 and 11 are suitable for the cross-linking of AZC with starch or cellulose, with an optimal pH level of 9. Javed et al. [21] selected a pH level of 10, and the pH of the starch–lignin–AZC solutions was adjusted using ammonia or sodium hydroxide. The reduction in the migration of lignin from films of starch–lignin–AZC blends was much more significant when the pH of the solutions was adjusted with ammonia than with sodium hydroxide. McAlpine [22] suggested that AZC and unmodified starch can interact with each other. A schematic illustration of the cross-linking of starch with AZC is shown in Figure 1. 

In the present study, the pH of the coating solutions was adjusted between 10 and 8.9, without the use of sodium hydroxide or ammonia. AZC can react with starch when water is removed from starch-based films/coatings [25]. William and Susan [26] and Wang et al. [27] have reported that both ammonia and carbon dioxide are released during drying and that the reactive sites on AZC react with the carbonyl and/or hydroxyl groups present on the polymer in a cross-linking reaction.

The presence of defects, such as pinholes or cracks, in a coating or film affects its barrier properties. Jamieson and Windle [28] have shown that the oxygen permeability of polymer films increased with the increasing number and diameter of pinholes present in the films. Greener et al. [29] modelled and studied the moisture permeation experimentally through multilayered barrier films. They proposed a relation between the thickness of the multilayered film and the size of pinholes. The presence of air bubbles in the coating colors, the coating procedures, and the drying strategies used can cause defects in the coating layers. However, most studies focused on the barrier properties of materials suitable for paper and paperboard coatings have been performed under laboratory conditions at low drying rates (e.g., by evaporation from petri dishes or by coating processes at low speed). In the present study, the coated papers were produced using a pilot coating machine operating at “industrial” speed (where the speed of the moving substrate was 400 m/min). Weigl and Grossmann [30] have investigated the runnability of coating colors at high machine speed and reported that problems such as streaking, skip coating, and bleeding can arise at high coater speeds. In general, skip coating was found to depend on the trapped air or gases in the coating color. Additives that lowered the surface tension of the wet coating color increased the risk of air entrapment and skip coating [30]. Rückert [31] explained that blade scratches and streaks are less common with rod metering units than with conventional blades, as the rotation of the rod opposite to the backing roll can hinder the particles from becoming stuck under the rod. 

Javed et al. [21] indicated that pH adjustment by ammonia reduced the amount of base in dry starch–lignin–AZC films which, in turn, would have a positive effect in reducing the migration from dry films to various aqueous solutions. In some industrial applications, the use of ammonia may be problematic, due to its low vapor pressure. Sodium hydroxide can sometimes be a better option. However, the limited solubility of kraft lignin requires that the pH is relatively high (i.e., requiring relatively high levels of bases, such as sodium hydroxide). Our hypothesis is to utilize the interaction between starch and lignin, in order to reduce the amount of base in the films, thereby reducing the migration of lignin from the dry film to surrounding aqueous solutions.

The present study has two main objectives. The first objective was to quantify how the migration of lignin to water and aqueous solutions is affected by the interaction between starch and kraft lignin and to assess the resulting possibility of decreasing the pH of coating solutions. The second objective was to study how up-scaling earlier laboratory results [20] to a pilot scale coating affects the properties of the coatings. Scaling up from laboratory coating trials can potentially cause defects, such as skin formation, blisters, pinholes, and cracks, due to the higher evaporation rates during the drying process. Pilot-coatings based on starch–lignin blends with and without AZC were applied to paper, using soft blade metering and smooth rod metering. Standardized pinhole tests based on red dye dissolved in ethanol, scanning electron microscopy (SEM), and a surface profiler were used to search for and quantify the defects in the coated paper samples caused by the faster drying rates in the pilot coater. The water vapor barrier properties of the pilot-coated paper, as well as the migration of lignin from the coatings into three model liquids were also examined.

## 2. Materials and Methods 

### 2.1. Materials

The starch, Emsol K 55-hydroxypropylated, and oxidized potato starch, was obtained from Emsland-Stärke GmbH, Emlichheim, Germany. The as-received starch was degraded to a molecular weight (*M*_w_) of ~1.5 × 10^6^ Da and the degree of hydroxypropylation of the starch was 0.1, according to the supplier. The viscosity of a 25 wt.% jet-cooked starch solution was 70 mPa at 70 °C, as determined by a Brookfield HAT at 50 rpm. The number of carboxyl groups per anhydrogluclose unit of the hydroxypropylated and oxidized potato starches of similar viscosity are typically in the range of 0.01–0.03 [32,33].

Kraft lignin, isolated by the LignoBoost process [5], was supplied by RISE LignoDemo AB, Kristinehamn, Sweden. The dry solids content and the pH of the as-received lignin solution were 27% and 11, respectively. The concentration of sodium ions in the lignin sample was 18.4 g/kg. The viscosity of the as-received lignin solution at 23 °C was about 300 mPa s, as measured by a Brookfield LVDV at 100 rpm.

Brenntag Nordic AB, Malmö, Sweden delivered glycerol (GLYCERIN VEG), with a purity of 99.5%, and ammonium zirconium carbonate (AZC; ZIRLINK) solution in water, which were used as a plasticizer and cross-linker, respectively. The solids content of the AZC solution was about 45%, according to the supplier.

MG paper with a grammage of 59.9 ± 0.3 g/m^2^, supplied by Nordic Paper, Kristinehamn, Sweden, was used as a substrate for the coating trials. 

### 2.2. Methods

Starch–lignin blends were prepared, according to the recipes shown in Table 1. In the laboratory trials, the starch was cooked in a boiling water bath for 60 min under continuous stirring at 400 rpm with a dispersion blade impeller. The temperature of the cooked starch solution was about 95 °C. The starch solution was added to the lignin solution under stirring at 100–300 rpm, followed by addition of glycerol and AZC solution according to the recipes shown in Table 1. The mixtures were cooled to room temperature prior to the addition of AZC. In the pilot trials, starch was cooked in a steam boiler for 60 min under vigorous stirring. The temperature of the cooked starch was 93 °C. The starch solution was added to the lignin solution under stirring at 300–400 rpm, followed by the addition of glycerol. The temperature of the mixtures was dropped to 68 °C. The mixtures were then kept under gentle stirring at room temperature for about 15 h, prior to the addition of AZC. The final temperatures of the mixtures for Recipes 1, 2, and 3 were 43, 35, and 31 °C, respectively. The solution containing starch, lignin, and AZC had a pH of 9.4, without further pH adjustment. The pH of the other solutions was adjusted using a 0.4 M HCl solution. The final pH, solids content, and viscosity of the solutions are shown in Table 1.

In order to test the lignin solubility with and without starch, starch–lignin, and lignin solutions were prepared, containing similar amounts of glycerol and AZC at different pH levels. The starch to lignin ratio was 70:30 on a dry basis. In the solubility tests, the final concentration of the mixtures was 15 wt%, and the pH of the solutions was adjusted with 0.4 M HCl solution. A Leica DMD108 microscope was used to detect precipitation.

The pilot-coated trials were performed at UMV Coating Systems AB, Säffle, Sweden, using the recipes shown in Table 1. One, two, three, and four coating layers were applied. The coated paper was dried before the next layer was applied. With one exception, the coatings were applied with an Invo-Coater using an Invo-Tip metering element (i.e., a soft blade) at different tip angles between 17° and 25°. The only exception was that a smooth rod metering element with a diameter of 14 mm and rotating at 40 rpm was used for Recipe 3 when the third and fourth coating layers were applied. The machine speed was 400 m/min. The coated paper was dried online in an Infra dryer, operating at 40%, 60%, and 40% of the full installed effect of 432 kW, followed by drying in three hot air dryers, at temperatures of 100–100–60, 100–100–60, and 150–150–60 °C for Recipes 1, 2, and 3, respectively. The base paper was also coated in laboratory-scale experiments, using a bench coater (K202 Control Coater, RK Coat Instruments Ltd., Royston, UK) equipped with a wire-wound rod (0.16 mm wire diameter), at a speed of 5 m/min. Four coating layers were applied, and the samples were dried at 105 °C for 90 s after each layer had been applied. The laboratory- and pilot-coated samples were kept at 23 °C and 50% RH (relative humidity) for at least 1 week before further testing.

The contact angle of water on coated paper was determined from images captured with a charge-coupled device camera (Basler acA1902–150um) over a time period of about 1 min using an OneAttention Theta Contact Angle Analyser (Biolin Scientific Oy, Espoo, Finland). The contact angle of water was plotted against time.

Scanning electron microscopy (SEM) was used to examine the defects, such as pinholes or pits, in the coatings. The samples were coated with gold in a sputter coater prior to testing. Micrographs were obtained in an SEM (FE-SEM, Leo 1530; Carl Zeiss GmBH, Vienna, Austria) under a low vacuum at 7 kV with secondary electron detection. The coated paper surface was also studied by Vertical Scanning Interferometry, using a surface profiler (Model 831-567-1, Bruker Nano GmbH, Berlin, Germany). The measurement type was vertical scanning interferometry (VSI), with a scanning speed of 1×. The measurements were made using a Mirau 50× objective with optical resolution of 0.5 µm, 0.55× field-of-view lens, green light, and a threshold frequency of 2%. The array size was 736× 480 pixels, where the pixel size was 0.35 µm.

Pinholes in the coating layers were detected according to SS-EN 13676 [34]. A colored solution was prepared by dissolving 0.5 g of dyestuff, Crossing Scarlet MOO (CAS 5413-75-2), in 100 mL of ethanol. A suitable amount of the solution was applied on the coated side of the samples and wiped off after 5 min. At least six specimens were tested, and the results are expressed as the mean number of pinholes/dm^2^ with a 95% confidence interval.

The water vapor transmission rate (WVTR) was measured, according to ISO 2528 [35], at 23 °C and 50% RH using the gravimetric dish method. The samples were mounted on a dish containing granular silica gel as a desiccant, and the change in weight over time was measured by weighing. There was a moisture gradient across the samples, due to the 50% RH outside and 0% RH inside the dish. The barrier-coated side of the sample was exposed to the 50% RH. Measurements were carried out in six replicates, and the mean values are presented with a 95% confidence interval.

The water absorptiveness (Cobb_60_ values) was measured, according to the SCAN-P 12:46 standard method, at 23 °C and 50% RH. The average values were calculated with a 95% confidence interval.

The dissolution of lignin from the coated samples immersed in water and aqueous solutions was tested. Pieces of coated paper containing 4 mg of the dry coating were put in 15 mL plastic test tubes containing 5 mL of deionized water, alkaline buffer with a pH of 10, or 3% *w/v* acetic acid solution (food simulant B) with a pH of 2.4. The exposed area of the coated paper (coated side only) was 4.9 ± 0.9 cm^2^. The tubes were rotated at 22 rpm, and the absorbance at 380 nm (Abs380) was measured using a UV-1800 240 V IVDD spectrophotometer (Shimadzu, Canby, OR, USA) after different times, in six replicates. The absorbance of lignin solutions was determined at various concentrations, in order to determine the amount of lignin that had migrated from the coated paper to the liquid phase. A wavelength of 380 nm was chosen, as Johansson et al. [14] have previously indicated that Abs380 has the highest selectivity for lignin. Starch, which may also migrate from the coated paper to the liquid phase, had a negligible influence on absorbance values at 380 nm. For the starch–lignin films (Recipe 1), the influence of starch on the absorbance values was 0.9% in deionized water, 0.7% in alkaline buffer solution, and 4% in food simulant B [21].

## 3. Results and Discussions

The water contact angle (WCA) on uncoated base paper and pilot-coated samples is presented in Figure 2. The presence of soluble starch made it possible to reach a relatively low pH (8.9) in the solution denoted by Recipe 3.

The WCA of the AZC-containing samples was substantially higher than that of the corresponding samples coated with starch–lignin formulations without AZC, which were similar to the WCA of the uncoated base paper. The reduction in water contact angle for papers coated with Recipe 1 was in good accordance with the reduction in WCA of paper and paperboard after surface treatment with starch reported elsewhere [33]. The WCA values obtained with Recipe 1 were rather similar to the WCA values for technical lignin, ranging from 40 to 60°, as reported by Borrega et al. [17]. Figure 2 clearly indicates that the cross-linking of the starch–lignin coatings by AZC made the surface more hydrophobic. It is probable that the effects of cross-linking on the WCA can be explained by the interaction between starch and AZC reported in the literature [22,23,27]. The increased molecular mobility of biopolymers (conformation changes and reorientation of side chains) may promote polymer–water interactions, resulting in low WCA [36]. The cross-linking between starch and AZC may have reduced the molecular mobility of starch and lignin present on the surface and, thereby, reduced the hydrophilicity of the coated paper surface. It is well-known that the hydroxyl groups located in the surface layer of starch molecules are slightly mobile and can rotate, to some extent. In contact with air (hydrophobic), the hydroxyl groups point inwards to the polymer matrix while, in contact with water (hydrophilic), the hydroxyl groups point outwards to the ambient water [37,38]. In addition, the almost constant WCA for Recipes 2 and 3 in Figure 2 indicated that cross-linking by AZC reduced the absorption of water into the coated papers. The higher pH of Recipe 2, compared to that of Recipe 3, did not affect the water absorption.

Lignin and starch–lignin solutions containing AZC and glycerol at different pH levels were examined with a microscope (Table 2), and precipitation was observed in the lignin solution when the pH of the lignin solution was reduced to 9.3 (Figure 3a). In the solutions containing starch, no precipitation was observed until the pH was reduced to 8.6. The observed starch–lignin solutions at pH 8.4 are shown in Figure 3b. Zhu et al. [39] have investigated the precipitation of lignin in a LignoBoost process, and found that lignin precipitation increased with decreasing pH. The increased solubility of lignin at low pH is clearly an effect of the presence of starch and indicates interactions (complex formation) between starch and lignin in solution. Richardson et al. [19] and Spiridon et al. [18] have reported that starch and lignin can interact with each other to form a complex, which probably made it possible to decrease the pH of the lignin solution containing starch. Interactions between starch and lignin seemed to reduce the concentration of “free” lignin molecules in the solution, compared to the corresponding solution without starch.

Figure 4 shows SEM micrographs of the laboratory- and pilot-coated samples when four coating layers were applied. Defects such as pinholes and pits were observed in all the pilot-coated samples, while no pinholes or pits were found in the laboratory-coated samples. However, due to the small area analyzed in each SEM image, this observation does not exclude the possibility of finding pinholes when larger areas are analyzed by other methods (such as the method in SS-EN 13676 [34]). However, it is evident that the number of pinholes or pits in the coating layers was greater when AZC was added to the coating formulation (see Figure 4d,e), when all the coating layers were applied using blade metering. The number of pinholes or pits in the AZC-containing coatings decreased significantly when the two top layers were applied using rod metering, rather than blade metering (see Figure 4e,f).

During pilot-coating, skip coating was observed with Recipe 3 after two coating layers had been applied using blade metering. Rod metering was, therefore, used in the application of the top layers, which hindered skip coating. Weigl and Grossmann [30] have reported that, in general, skip coating depends on trapped air or gases in the coating color and that additives lowering the surface tension of the wet coating color increase the risk of air entrapment and, therefore, skip coating. Lignin is rather hydrophobic in nature [1]. Delgado et al. [40] have shown that dissolved alkali kraft lignin (M_w_ ~1.5 10^4^ Da) substantially decreased the surface tension of water, where the decrease in surface tension was even stronger if fractionated alkali lignin of lower M_w_ was used. Bylin et al. [41] found that the M_w_ of LignoBoost lignin was ~1.6 10^4^ Da (i.e., similar to the alkali kraft lignin used by Delgado et al. [40]). The hydrophobic nature of lignin decreased the surface tension when dissolved in water, compared to the surface tension of pure water. This property of kraft lignin may, thus, increase the tendency for skip coating to occur.

Cracks in the SEM images were observed in all laboratory- and pilot-coated samples (Figure 4). It seemed that the cracks in the coating layers might have appeared during the SEM procedure, due to the evaporation of water under the low vacuum pressure, as no cracks were detected in any of the coating layers when the samples were studied using a surface profiler (Figure 5). Surface profiler analyses, such as that shown in Figure 5, have been proven to be useful for the detection of narrow cracks in coated paperboards [42]. The resolution of the surface profiler images made it possible to detect cracks with a width of just a few micrometers, if they would have existed. In the Y-profile of the image taken by a surface profiler, the depth of the pinhole on the right-hand side of the image is about 12 µm and is about 5 µm on the left-hand side. The total thickness of the four coating layers was about 10 µm, which indicates that one of the pinholes or pits went through all the coating layers, while the others were only in the top coating layers. One possible explanation is that the transport of moisture from the base-paper and previous coating layers followed already existing pinholes or pits during the rapid evaporation in the pilot trials. It has been reported that ammonia and carbon dioxide are released during the drying of polymer coatings containing AZC [22,26,27]. The evolution of ammonia and carbon dioxide could also cause defects, such as pinholes, in the coatings during drying.

Table 3 shows the coat weight, the number of pinholes in the coating layers, the water vapor transmission rate (WVTR) through pilot-coated paper, the water absorptiveness (Cobb_60_), and the surface roughness, when all the coating layers were applied using blade metering, except for the top two layers in Recipe 3, which were applied using rod metering. No pinholes were observed in the samples coated in the laboratory with starch/lignin blends without AZC. The number of pinholes in the pilot-coated samples was high (i.e., greater than 2000 pinholes per m^2^) after applying two and three coating layers for Recipe 1 without AZC and Recipe 2 with AZC, respectively. The number of pinholes in the laboratory-coated AZC-containing samples was much less than that in the corresponding pilot-coated samples, indicating that the rapid material and heat transport in the pilot coating process favored the formation of pinholes.

When four layers were applied in the pilot-scale coater, the number of pinholes was still rather high for all pilot-coated samples, even though a substantial reduction in the number of pinholes was observed for Recipe 1 without AZC. The addition of AZC to the recipes resulted in a larger number of pinholes in both the laboratory- and pilot-coated samples. The results in Figure 4, Figure 5, and Table 3 indicate that the ammonia and carbon dioxide developed during the drying of the AZC-containing formulations may have made the film covering pinholes or pits located underneath more susceptible to breaking up during the drying. Rückert [31] has stated that blade scratches and streaks are less common with roll blades (rod metering units) than with conventional blades (blade metering units) and explained that the rotation of the rod opposite to the backing roll in the pilot-coating unit could hinder particles from fastening and can partially remove air bubbles present in the coating color. This may have resulted in fewer air bubbles (and, thus, fewer pinholes or pits) when roll metering was used for the top coating layers based on the formulation containing AZC.

The WVTR of the pilot-coated papers was generally high for all the samples, even after four coating layers had been applied. There was, however, a significant decrease in the WVTR for the pilot-coated papers, even when a single coating layer was applied, despite the fact that the coating layer contained a large number of pinholes. The WTVR for the pilot-coated samples decreased with the number of coating layers. The addition of AZC to the formulation generally resulted in slightly higher WVTR values and higher Cobb_60_ values for the AZC-containing samples (Table 3). This indicates that both the uptake of liquid water and water moisture are promoted by the addition of AZC, although the influence of contact time on WCA (Figure 2) indicated that AZC decreased the hydrophilic nature of the surfaces. One possible explanation is that the uptake of liquid water and water moisture is controlled by surface defects such as pinholes. The lower pH of Recipe 3, compared to Recipe 2, did not result in any significant difference in WVTR between the two formulations. Table 3 shows no significant differences in roughness of the coated paper between the three coating solutions. Thus, the differences in WCA shown in Figure 2 cannot be explained in terms of roughness differences; however, WCA for the corresponding perfect flat surfaces may be slightly higher than presented in Figure 2, especially for the sample coated with Recipe 1, with the highest deviation from 90°.

The amount of lignin that migrated to the three model liquids from the pilot-coated papers at different pH levels after different contact times is shown in Figure 6, Figure 7, and Figure 8. 

The model liquids used were deionized water with a pH around neutral, alkaline buffer solution with pH 10, and food simulant B (3% *w/v* acidic acid solution) with pH 2.4. The amount of lignin that migrated from the starch–lignin–AZC-based coatings into the model liquids at a given time was lower than that from the starch–lignin-based coatings without AZC. A similar effect of the addition of AZC on the migration of lignin from corresponding self-supporting films has been observed previously, where the solutions with and without AZC were adjusted to pH 10 with ammonia [21]. In the previous study, 20 mg of films with an exposed area of ca. 2 cm^2^ were tested, while in the present study, the samples had 4 mg of coatings with an exposed area of 5 cm^2^. Javed et al. [21] showed that the migration of lignin from the films was substantially lower when the pH of the solution was adjusted with ammonia, compared to neutralization with sodium hydroxide. 

The addition of ammonia solution to adjust the pH of the starch–lignin–AZC solutions enhanced the interaction between AZC and starch [21]. The results presented in Figure 6, Figure 7, and Figure 8 indicate that the reduction in pH of the solutions to pH = 8.9 led to a similar level of lignin migration as neutralization by ammonia at a pH of 10. Moreover, the migration of lignin from the coatings in contact with the aqueous phase of food simulant B with low pH was significantly lower than that from the coatings in contact with deionized water or alkaline buffer solution at high pH. The decrease in pH reduced the migration of lignin from the coatings, due to the low solubility of lignin at low pH. The results indicated that both methods—i.e., (i) neutralization by ammonia and (ii) lowering the pH of starch-lignin solutions containing AZC—can be used to improve the stability of the films/coatings in water. The decrease in lignin migration when AZC was added to the recipe can be explained by the formation of complexes between starch and lignin and the cross-linking of AZC with starch [18,22].

The present study highlighted the pros and cons of cross-linking in starch/lignin solutions by AZC. The advantages were related to the increased stability in water (Figure 6, Figure 7, and Figure 8) and a more hydrophobic surface, in terms of the water contact angle (Figure 2), while the disadvantages were related to the increased number of pinholes in both laboratory- and pilot-coated paper (Table 3). As the addition of AZC apparently leads to properties that can be regarded as both negative and positive, it is likely that the cross-linking of AZC is not suitable for all packaging applications. As some of the effects of AZC seem to be contradictory, an optimal amount of AZC may exist, in order to optimize the overall properties of the coatings for a specific packaging application.

## 4. Conclusions

In this study, the effect of different pH levels on lignin solubility in the presence of starch and on the migration of lignin from starch–lignin coatings with and without AZC were investigated. The water stability of starch–lignin blends, measured through the migration of lignin to ambient water and aqueous solutions, can be improved considerably with the addition of AZC as a cross-linker. In addition, AZC made the surface more hydrophobic, in terms of increases in the water contact angle.

Solubility of lignin in water was required for the preparation of the coating solutions. Aqueous solutions of lignin can be prepared at lower pH by blending lignin and starch, due to interactions between starch and lignin. The addition of starch made it possible to reduce the pH of the coating solution from 9.4 to 8.9 which, in turn, reduced the water solubility of lignin in the dry coatings. 

With respect to the formation of defects that may occur in the coating process, the addition of AZC seemed to increase the number of pinholes in the pilot-coated papers. However, the design of the pilot coating can be used to minimize the number of pinholes. Compared to the uncoated paper, the WVTR of the pilot-coated papers was decreased significantly, even when a single coating layer was applied, although the WVTR values were generally high for all the pilot-coated samples. Lowering the pH from 9.4 to 8.9 had no significant effect on the water contact angle or the WVTR of the coated samples.

## Figures and Tables

**Figure 1 polymers-13-01595-f001:**
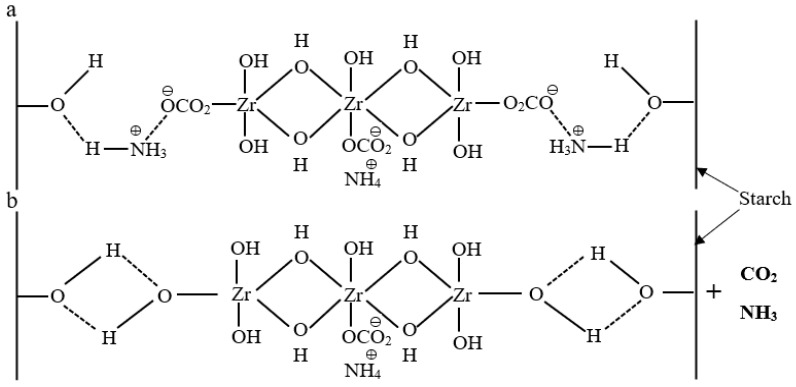
The reaction of ammonium zirconium carbonate (AZC) with starch: (**a**) AZC reaction with starch through hydrogen bonding and (**b**) release of NH_3_ and CO_2_ upon drying.

**Figure 2 polymers-13-01595-f002:**
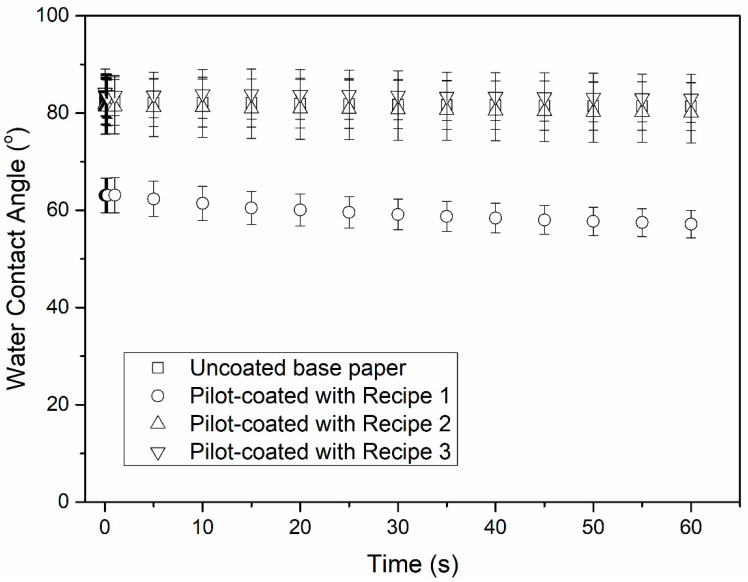
Contact angle of water on the uncoated and pilot-coated samples with starch/lignin blends using Recipes 1, 2, and 3, at 23 °C and 50% RH (relative humidity).

**Figure 3 polymers-13-01595-f003:**
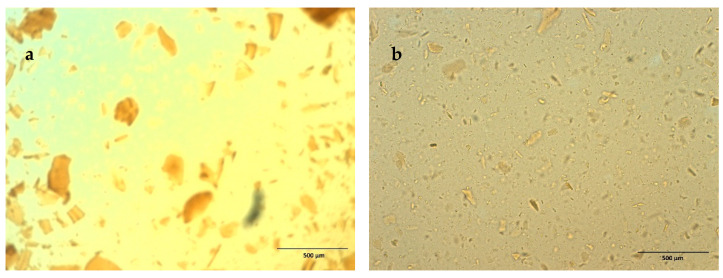
Light optical micrographs showing precipitation in: (**a**) lignin solution at pH 9.3 and (**b**) starch–lignin solution at pH 8.4.

**Figure 4 polymers-13-01595-f004:**
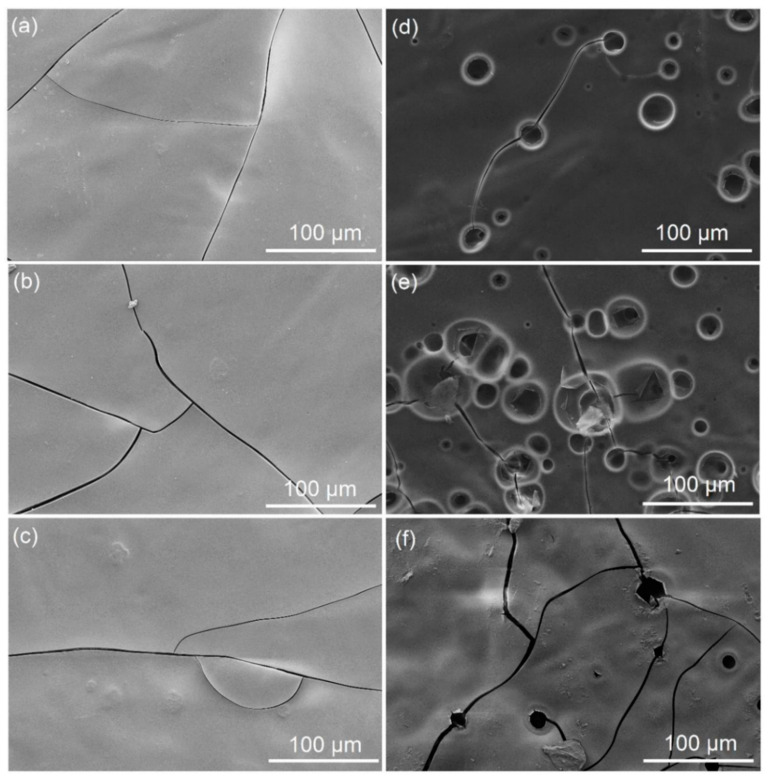
SEM images of laboratory-coated (**a**–**c**) and pilot-coated (**d**–**f**) samples with starch/lignin blends when four coating layers were applied: (**a**,**d**) without AZC at pH = 10, (**b**,**e**) with AZC at pH = 9.4, and (**c**,**f**) with AZC at pH = 8.9. Coated by blade metering for all layers (**d**,**e**) or blade metering for layers 1 and 2 and by rod metering for layers 3 and 4 (**f**).

**Figure 5 polymers-13-01595-f005:**
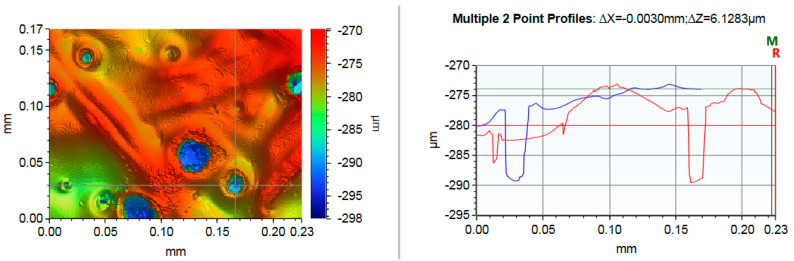
Surface profile image of a pilot-coated sample with starch/lignin blends without AZC when four coating layers were applied.

**Figure 6 polymers-13-01595-f006:**
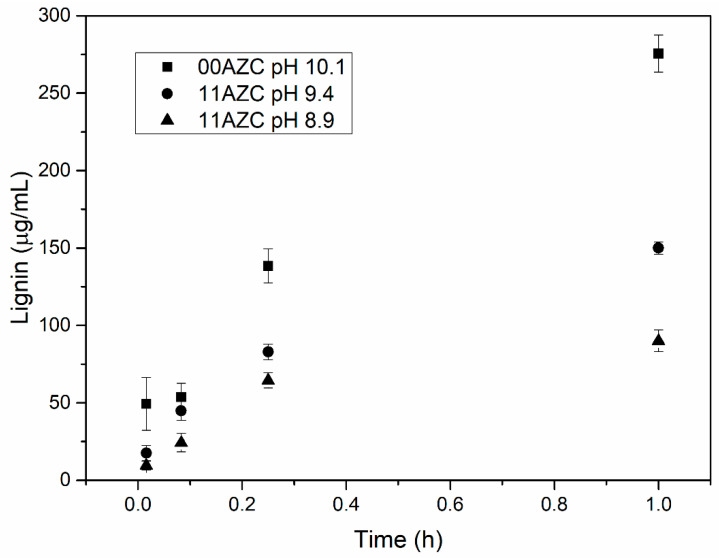
Migration of lignin from the pilot-coated paper to deionized water after different times.

**Figure 7 polymers-13-01595-f007:**
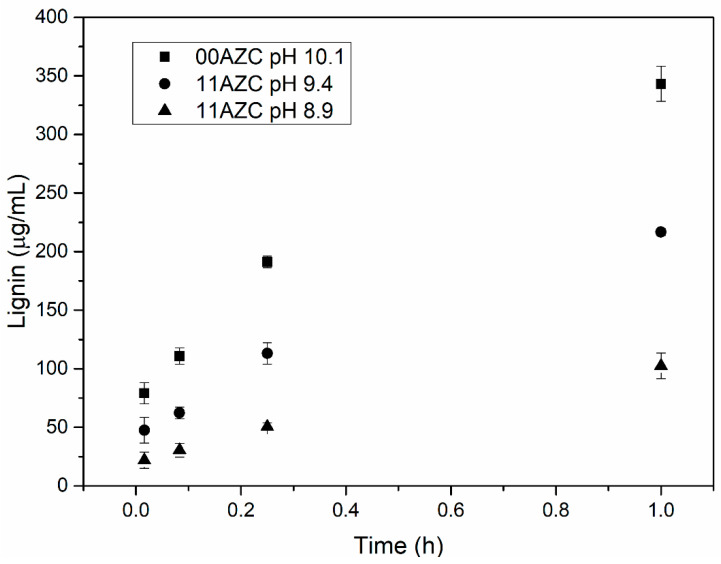
Migration of lignin from the pilot-coated paper to an alkaline buffer solution after different times.

**Figure 8 polymers-13-01595-f008:**
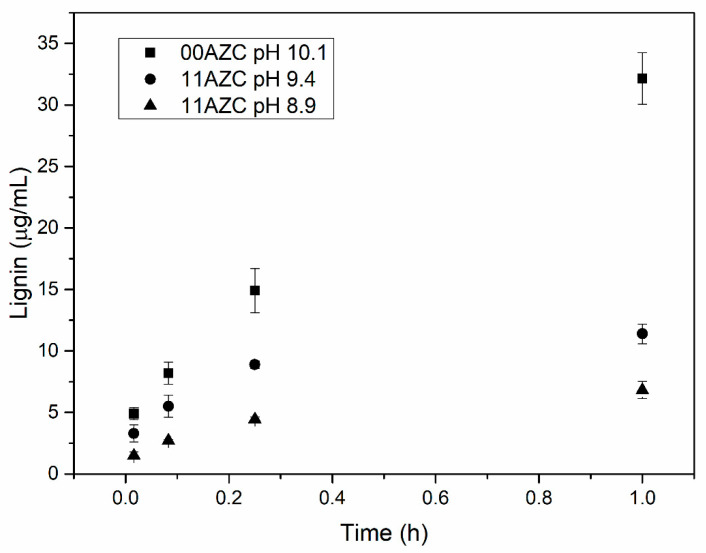
Migration of lignin from the pilot-coated paper to a 3% (*w/v*) acetic acid solution after different times.

**Table 1 polymers-13-01595-t001:** Recipes of starch, lignin, glycerol, and ammonium zirconium carbonate (AZC) blends, as well as the pH, solids content, and viscosity values of the final mixtures at 23 °C used for coating the paper substrate.

Recipe	Starch	Lignin	Glycerol	AZC	pH	Solids Content	Viscosity
	Weight (%)					(%)	(M Pa s)
1	62	19	19	0	10.0	31.5 ± 0.8	320
2	55	17	17	11	9.4	30.2 ± 0.4	780
3	55	17	17	11	8.9	29.7 ± 0.9	630

**Table 2 polymers-13-01595-t002:** Solubility of lignin with and without starch, containing similar amounts of AZC and glycerol, at different pH levels. All the solutions contained AZC and were at 23 °C.

pH	Lignin	Starch–Lignin
9.3	Precipitation observed	No precipitation
9.2	Precipitation observed	No precipitation
9.0	–	No precipitation
8.8	–	No precipitation
8.6	–	No precipitation
8.4	–	Precipitation observed

**Table 3 polymers-13-01595-t003:** Coat weight, number of pinholes (according to SS-EN 13676 [34]), water vapor transmission rate (WVTR), water absorptiveness (Cobb_60_), and roughness values (Ra) of the coated paper. All the pilot-coated layers were applied using blade metering, except for the top two coating layers of Recipe 3, which were applied using rod metering.

Recipe	Number of Layers		Coat Weight (g/m^2^)		Pinholes (Number/m^2^)		WVTR	Cobb_60_	Roughness Values (Ra)
							(g/m^2^.d)	(g/m^2^)	(µm)
	Laboratory	Pilot	Laboratory	Pilot	Laboratory	Pilot	Pilot	Pilot	Pilot
Base paper	–	–		–		>2000	479 ± 5	80 ± 3	–
1	–	1	–	1.9 ± 0.4	–	>2000	248 ± 5	–	–
	–	2	–	4.1 ± 0.4	–	>2000	187 ± 4	–	–
	–	3	–	6.8 ± 0.4	–	415 ± 133	131 ± 3	–	–
	4	4	14.3 ± 1.5	9.4 ± 0.5	0	109 ± 58	119 ± 2	92 ± 15	1.2 ± 0.1
2	–	1	–	2.2 ± 0.2	–	>2000	278 ± 4	–	–
	–	2	–	4.4 ± 0.3	–	>2000	236 ± 10	–	–
	–	3	–	7.0 ± 0.3	–	>2000	167 ± 5	–	–
	4	4	15.4 ± 1.3	9.7 ± 0.3	85 ± 29	1334 ± 518	166 ± 17	162 ± 3	1.6 ± 0.6
3	–	1	–	2.3 ± 0.2	–	>2000	289 ± 8	–	–
	–	2	–	4.5 ± 0.3	–	>2000	234 ± 6	–	–
	–	3	–	7.5 ± 0.3	–	1311 ± 560	149 ± 9	–	–
	4	4	14.4 ± 1.3	9.9 ± 0.4	76 ± 41	751 ± 192	155 ± 6	141 ± 15	1.4 ± 0.3

## Data Availability

The data presented in this study are available on request from the corresponding author.

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
