# Peer review of "Lignin-Containing Coatings for Packaging Materials—Pilot Trials"

_polymers, 2021, doi:10.3390/polym13101595_

Round 1

Reviewer 1 Report

This work presents "Lignin-containing coatings for packaging materials-pilot trials”. Developing new bio-based coating formulations for packaging materials is an important issues for both academic research areas and industries. This manuscript is recommended to be published after including and addressing the below listed comments with major corrections.

- The authors should eliminate the current grammatical and punctuation mark errors and also confirm the correct scientific English.

- The authors should clearly explain the innovation and importance of their work on the introduction part of the manuscript. The problem is not well presented in the introduction and the objectives are not well identified.

- The authors should justify the value of the work and compare their work with previously similar published papers. Some irrelevant information in the introduction –especially in the first paragraph- must be eliminated. Instead barrier coating can be included. The introduction section needs to be elaborated.

- The authors should work on the scientific English of the manuscript and elaborate it.

- Why did the authors not consider to measure oxygen permeation as well? This is as important as water vapor permeation for barrier coatings

- The starch-lignin interaction is claimed to be a key factor and affected the pH of precipitation and solubility of lignin. It is suggested that, the authors confirm the interaction between starch and lignin using chemical analysis methods.

- The authors provided the surface profile of the samples in Fig. 4. However, since surface roughness effect the hydrophilic properties of the material, I recommend the authors to compare the surface roughness of the papers coated with different recipes.

- Although the water contact angle test provides some information about the hydrophilic properties of coated papers, however, for better understanding the effect of coating, I suggest the authors to conduct another common sizing degree evaluation method to compare the effect of coating.  For example using a high grammage papers as base paper and conduct a cobb test.

- Why did the authors use different drying temperatures which can affect the consolidation and migration of coating ingredients and thereby the properties of coated samples. For example, Figure 3, shows larger crack area for the pilot coated samples which might be attributed to higher drying temperature.

- The authors claimed that the cracks in the SEM images might have appeared during the SEM procedure. I suggest authors to evaluate this by optical microscopy.

- Please check the weight percent of coating ingredients in Table 1.

- In the Methods section, line 137 : “A microscope, Leica DMD108, was used to detect precipitation” . Please provide the images.

In case of starch-lignin sample at pH value of 8.6, how long after mixing the ingredients the precipitation was occurred? How about the stability of the coating formulations without precipitation.

- Instead of Fig. 1, which was presented in their previous work (Ref. 16), the authors may provide the Schematic illustration of the crosslinking of starch with lignin by AZC.

- The authors should write the complete terms of all abbreviations (including the instruments) before the first use in the abstract and main manuscript.

- Some parts of the manuscript are confusing or not clear for the readers. For example:

  • Line 13-14: “The effect on the solubility of lignin of the pH level with and without ammonium zirconium carbonate was examined”
  • In Line 238: “Defects such as pinholes and pits were observed in all the pilot-coated samples, but no pinholes or pits were found in the lab-coated samples.” However, in Line 291-292 and Table 3, the authors are mentioning about the pinholes in both lab- and pilot- coated samples: “The addition of AZC to the recipes resulted in a larger number of pinholes in both the laboratory- and pilot-coated samples”. The authors may pick SEM images which show existence of pinholes in lab-coated papers.
  • In the Conclusion section, Line 357-358 the authors mentioned about the optimum amount of AZC. However, in this research only one level of AZC was applied.

Author Response

We would like to appreciate the reviewers valuable input given to our manuscript. We have considered all the comments, concerns and suggestions given by the reviewers. We have corrected our manuscript accordingly. Now we believe this revised manuscript meets the standards of the journal and expectations of the reviewers as well as the readers of the journal. The response to the each and every comments of the reviewers is given below:

Comment: This work presents "Lignin-containing coatings for packaging materials-pilot trials”. Developing new bio-based coating formulations for packaging materials is an important issues for both academic research areas and industries. This manuscript is recommended to be published after including and addressing the below listed comments with major corrections.

Response: First, we want to express our gratitude to the reviewer for the constructive comments. The comments were essential to remove all flaws from our manuscript. We hope that we have succeeded to incorporate all proposals in our revised manuscript.

Comment: The authors should eliminate the current grammatical and punctuation mark errors and also confirm the correct scientific English.

Response: Thank you for the suggestion, we have corrected grammatical and punctuation mark errors and enhanced the scientific English terminologies relevant to this research domain.

Comment: The authors should clearly explain the innovation and importance of their work on the introduction part of the manuscript. The problem is not well presented in the introduction and the objectives are not well identified.

Response: We have now reformulated this section and clearly identified our two main objectives and hypothesis (line 119-132). In addition we have revised the section “Conclusions” in order to highlight the links between our objectives and our findings (line 422-440).

Comment: The authors should justify the value of the work and compare their work with previously similar published papers. Some irrelevant information in the introduction –especially in the first paragraph- must be eliminated. Instead barrier coating can be included. The introduction section needs to be elaborated.

Response: A brief summary of literature review and comparison is presented in the introduction part of the manuscript. New references are added to the introduction (line 49-60). More elaborations and summarized information is also added (line 97-101). We believe that the first section now gives a relevant background for this manuscript.

Comment: The authors should work on the scientific English of the manuscript and elaborate it.

Response: See our response to the first comment above.

Comment: Why did the authors not consider to measure oxygen permeation as well? This is as important as water vapor permeation for barrier coatings.

Response: One of the objectives of this paper was to study upscaling to pilot scale, i.e. coating at high machine speeds. Pilot coating is very challenging, and very few reports exist that have succeeded to get coated papers or coated paperboards free of defects. The SEM and surface profiler images of the pilot-coated samples reveals that coatings contain defects such as pinholes Fig.4. We also have performed standard pinhole test according to SS-EN 13676 standard to detect pinholes in the coatings. Results presented in Table 3 of this paper shows that all the pilot coated samples contain a large number of pinholes. For such number of pinholes, it is obvious that OTR-values for these coated samples would be over the measurable range i.e., >10,000 cm3/(m2 day), when measured according to the ASTM D 3985-05 standard using a Mocon Ox-Tran oxygen transmission rate tester. Therefore, we believe that it is not worthwhile to measure oxygen permeation.

Comment: The starch-lignin interaction is claimed to be a key factor and affected the pH of precipitation and solubility of lignin. It is suggested that, the authors confirm the interaction between starch and lignin using chemical analysis methods.

Response: Researches have already confirmed the interaction between starch and lignin. The findings of researchers along with the references are now added in the introduction section (line 49-60). Therefore, we believe that the confirmation of starch and lignin interactions using chemical analysis methods is out of the scope of the study. However, results from the solubility tests (Table 2) clearly indicates interactions between starch and lignin. In several research papers, indirect methods such as rheology (changes in flow properties and storage modulus) are used in order to confirm interaction. To some extent, solubility may also be such indirect method.

Comment: The authors provided the surface profile of the samples in Fig. 4. However, since surface roughness effect the hydrophilic properties of the material, I recommend the authors to compare the surface roughness of the papers coated with different recipes.

Response: Roughness values (Ra) for different recipes are now added in the Table 3. Their comparison is also described in the text (line 373-377).

Comment: Although the water contact angle test provides some information about the hydrophilic properties of coated papers, however, for better understanding the effect of coating, I suggest the authors to conduct another common sizing degree evaluation method to compare the effect of coating.  For example using a high grammage papers as base paper and conduct a Cobb test.

Response: Thank you for the suggestion, we have added Cobb­60 results in Table 3 and compared (line 366-371).

Comment: Why did the authors use different drying temperatures which can affect the consolidation and migration of coating ingredients and thereby the properties of coated samples. For example, Figure 3, shows larger crack area for the pilot coated samples which might be attributed to higher drying temperature.

Response: The cracks seen in SEM images for laboratory and pilot coated samples are formed due to the vacuum applied during SEM analysis. The samples were observed to have no cracks when analyzed using the surface profiler Fig. 5.

Comment: The authors claimed that the cracks in the SEM images might have appeared during the SEM procedure. I suggest authors to evaluate this by optical microscopy.

Response: We have evaluated the sample with surface profiler and we have found cracks in the coated samples tested. We have also added more information about the surface profiler analysis in methods sections (line 204-208), so it will be evident for a reader that the resolution of the surface profiler is sufficient in order to detect narrow cracks as those visible in the SEM micrographs. We believe that the surface profiler is capable in identifying cracks in coated sample (line 312-315).

Comment: Please check the weight percent of coating ingredients in Table 1.

Response: Thank you for the correction, we have rechecked and corrected the weight percentage of the coating ingredients Table 1.

Comment: In the Methods section, line 137: “A microscope, Leica DMD108, was used to detect precipitation” . Please provide the images.

Response: Images are included Fig. 3.

Comment: In case of starch-lignin sample at pH value of 8.6, how long after mixing the ingredients the precipitation was occurred? How about the stability of the coating formulations without precipitation.

Response: We have added optical microscope images for the detected precipitation to the revised manuscript Fig. 3. The precipitation occurred immediately after pH adjustments.

Comment: Instead of Fig. 1, which was presented in their previous work (Ref. 16), the authors may provide the Schematic illustration of the crosslinking of starch with lignin by AZC.

Response: Fig. 1 is changed.

Comment: The authors should write the complete terms of all abbreviations (including the instruments) before the first use in the abstract and main manuscript.

Response: We have defined all the abbreviation used in the abstract as well as in the main manuscript, line 14, 15.

Comment: Some parts of the manuscript are confusing or not clear for the readers. For example:

  • Line 13-14: “The effect on the solubility of lignin of the pH level with and without ammonium zirconium carbonate was examined”

Response: Revised “The effect of pH level on the solubility of lignin with and without ammonium zirconium carbonate was examined”.

  • In Line 238: “Defects such as pinholes and pits were observed in all the pilot-coated samples, but no pinholes or pits were found in the lab-coated samples.” However, in Line 291-292 and Table 3, the authors are mentioning about the pinholes in both lab- and pilot- coated samples: “The addition of AZC to the recipes resulted in a larger number of pinholes in both the laboratory- and pilot-coated samples”. The authors may pick SEM images which show existence of pinholes in lab-coated papers.

Response: Pinholes in Table 3 were detected with a standard method SS-EN 13676 with a staining liquid over five specimens of > 1 dm2. The analyzed area is one dm2. The SEM images were used illustrating the microstructure of the surface. We have added that pinholes were determined according to SS-EN 13676 in the table description text. We believe that it is difficult to use SEM images to get statistically significant results on the existence of pinholes. The method SS-EN 13676 gives results, which are statistically significant. We have also reformulated line 287-289.

  • In the Conclusion section, Line 357-358 the authors mentioned about the optimum amount of AZC. However, in this research only one level of AZC was applied.

Response: We have revised this sentence and moved it to the discussion section, since this is merely a discussion than a conclusion (line 413-421).

Reviewer 2 Report

The authors describe the formulation of lignin/starch/AZC, their solubility at different pH, and coating characteristics. The experimental part as well as the results section are well performed and reported, supporting the conclusions.

I suggest publication in the present form, after careful proofreading.

Author Response

Dear Reviewer,

Thank you for your recommendation for publication of our manuscript.

Round 2

Reviewer 1 Report

Thanks for the revised version, which has materially improved the manuscript. I think this is ready to publish.